# Effect of Combining Wuyiencin and Pyrimethanil on Controlling Grape Gray Mold and Delaying Resistance Development in *Botrytis cinerea*

**DOI:** 10.3390/microorganisms12071383

**Published:** 2024-07-08

**Authors:** Jiabei Xie, Boya Li, Jia Li, Kecheng Zhang, Longxian Ran, Beibei Ge

**Affiliations:** 1State Key Laboratory of Biology of Plant Diseases and Insect Pests, Institute of Plant Protection, Chinese Academy of Agricultural Sciences, Beijing 100093, China; xiejiabei1998@163.com (J.X.); l18703242721@163.com (B.L.); kczhang@ippcaas.cn (K.Z.); 2College of Forestry Sciences, Hebei Agricultural University, Baoding 071000, China; longxianran@163.com; 3State Key Laboratory of Biocatalysis and Enzyme Engineering, School of Life Sciences, Hubei University, Wuhan 430062, China; lijia@hubei.edu.cn

**Keywords:** delay resistance, fungicide, compound mixture

## Abstract

By screening the compounding combination of Wuyiencin and chemical agents, this study aims to delay the emergence of chemical agent resistance, and provide a technical reference for scientific and rational fungicides technology. This study investigated the impacts of the antibiotic wuyiencin derived from *Streptomyces albulus* var. *wuyiensis* and its combination with pyrimethanil on the inhibition of *Botrytis cinerea*. Treatment with wuyiencin (≥80 µg mL^−1^) strongly inhibited the pathogenicity of *B. cinerea* and activated the plant defense response against *B. cinerea*. Application of 80–100 µg mL^−1^ wuyiencin effectively controlled grape gray mold (by 57.6–88.1% on leaves and 46.7–96.6% on fruits). Consequently, the application of 80–100 µg mL^−1^ wuyiencin effectively mitigated grape gray mold incidence, leading to a substantial reduction in disease symptoms to nearly imperceptible levels. When wuyiencin (at the median effective concentration [EC_50_]) was combined with pyrimethanil (EC_50_) at a ratio of 7:3, it exhibited the highest efficacy in inhibiting *B. cinerea* growth. This combination was significantly more potent (*p* < 0.05) than using wuyiencin or pyrimethanil alone in controlling gray mold on grape leaves and fruits. Furthermore, the combination effectively delayed resistance development in gray mold. The experimental results show that wuyiencin can delay resistance development by affecting the expression of methionine biosynthesis genes and reducing the activity of the cell wall-degrading enzyme activity.

## 1. Introduction

Grape (*Vitis vinifera* L.) is a fruit crop with the second highest production worldwide; it is a widely cultivated crop in China, with high cultural value [1]. There are more than forty known diseases affecting grapes, mainly gray mold, downy mildew, and powdery mildew. Gray mold, caused by *Botrytis cinerea* Pers., severely affects crop growth and yield, causing huge economic losses [2,3]. *B. cinerea* mainly overwinters in the host body or soil with mycelium or sclerotium, and the overwintered mycelium or sclerotium germinates under a suitable environment and produces conidia to infect the grape flower ear or young fruit. New conidia are quickly produced on the host tissue after the initial infection and continue to infect again; therefore, it is virtually impossible to control gray mold with a single control measure [4]. Presently, to prevent and control gray mold, a comprehensive strategy based on chemical control in combination with epidemic prediction, early diagnosis, agricultural control, and biological control is employed. The chemical fungicides used to control gray mold are categorized into four classes: benzimidazoles (e.g., carbendazim and thiophanate-methyl), dicarboximides (e.g., iprodione, procymidone, and vinclozolin), anilinopyrimidines (e.g., cyprodinil, mepanipyrim, and pyrimethanil), and N-phenylcarbamates (e.g., diethofencarb) [5]. Most target sites of chemical fungicides are specific; therefore, repeated use of these fungicides leads to resistance development—an increasingly serious problem—resulting in reduced disease control. It has been documented that *B. cinerea* has developed resistance to several chemical fungicides used to control gray mold [6,7,8,9,10].

Fungicide resistance has been documented for several chemical agents; garnering significant research attention. Analyzing the mechanisms of action of chemical fungicides is important to understand and effectively reduce or retard the resistance to fungicides [11,12]. For example, the mechanism of resistance to boscalid (a succinate dehydrogenase inhibitor) is attributed to point mutations in the succinate dehydrogenase genes *sdhB*, *sdhC*, and *sdhD* in fungal pathogens [13,14,15,16]. Studies have shown that long-term use of fludioxonil leads to a mutation in the histidine kinase gene osmosensing-1 (*OS-1*), which affects the histidine kinases/adenylyl cyclases/methyl-binding proteins/phosphatases (HAMP) domain, leading to the development of fludioxonil resistance in pathogens [17]. Chemical fungicides achieve rapid and effective disease control, but they need to be combined with other control methods to mitigate fungicide resistance [18].

In contrast, biocontrol agents act directly on pathogens, inhibiting their development, and stimulating host plant defense responses, thus achieving effective disease control. Biocontrol agents contain a mixture of active metabolites that target multiple sites in the pathogen genome; thus, the use of biocontrol agents does not easily induce fungicide resistance [19]. According to worldwide reports, various biocontrol agents such as *Trichoderma* spp. (e.g., *Trichoderma viride* TH4), *Bacillus* spp. (e.g., *Bacillus velezensis* NH-1 and *B. subtilis*), *Pseudomonas* spp. (e.g., *Pseudomonas aeruginosa* VIH2), *Streptomyces* spp. (e.g., *Streptomyces rectiviolaceus* DY46), and Saccharomycete (e.g., *Kloeckera apiculata* 34-9) have good control effects on gray mold [20,21,22,23,24,25,26,27,28].

Due to fungicide resistance and environmental contamination resulting from the chemical control of plant diseases, the use of a biocontrol agent in combination with chemical fungicides has been proposed; it effectively lowers disease incidence, reduces chemical residue on agricultural products, reduces environmental contamination, and mitigates fungicide resistance of the pathogen. This approach may also alleviate the problems of variable efficiency and slow action associated with biocontrol agents, thus achieving higher efficiency of plant disease management in a more environmentally/ecologically friendly manner. It has been documented that biocontrol agents compounded with chemical fungicides achieve effective control of fungal diseases. For example, a compound preparation of *Pseudomonas fluorescens* with thiabendazole significantly reduced apple gray mold [29], and that of *P. syringae* MA-4 with cyprodinil effectively controlled both blue and gray mold of apple [30]. In addition, the compound preparation of *B. subtilis* HMB-20428 and azoxystrobin can effectively prevent and control grape downy mildew [31].

Wuyiencin is a secondary metabolite produced by *Streptomyces albulus* var. *wuyiensis* and has strong antagonistic activity against various plant pathogens. It has been shown to effectively control cucumber powdery mildew (*Sphaerotheca fuliginea*), tomato gray mold (*B. cinerea*), soybean stem rot (*Sclerotinia sclerotiorum*), tea leaf blight (*Colletotrichum camelliae*), and gray blight (*Pestalotiopsis theae*) [32,33,34,35]. However, to date, there has been no systematic and comprehensive study on controlling grape gray mold caused by *B. cinerea* using wuyiencin. In the present study, the inhibitory effects of wuyiencin alone and in combination with some chemical fungicides on grape gray mold and its causative pathogen *B. cinerea* were investigated to identify an efficient and environmentally friendly strategy to control this disease.

## 2. Materials and Methods

### 2.1. Materials

*B. cinerea* strains B05.10, 57, 59, 61, 65, 503, and 514 were obtained from the Institute of Plant Protection, Chinese Academy of Agricultural Sciences (IPP-CAAS). They were cultured on potato dextrose agar (PDA) at 25 °C for 5 days to prepare inocula (mycelial discs or conidial suspensions) for use in different experiments. The grape cultivars Rose-scent (for seedling samples) and Fujiminori (for fruit samples) were grown in a greenhouse at the IPP-CAAS.

Plant material: Tender leaves of *Vitis vinifer* Muscat Hambuige (11 cm × 13 cm), moderately susceptible. Mature market *Vitis labrusca* Fujiminori grape fruits.

Potato dextrose agar (PDA): Potato starch, 4.0 g; dextrose, 20.0 g; and Agar, 15.0 g, were purchased from Becton, Dickinson and Company (Franklin Lakes, NJ, USA).

Fresh wuyiencin solution (800 µg mL^−1^) was obtained from the IPP-CAAS Biocontrol Research Laboratory. The chemical fungicides—40% pyrimethanil, 50% iprodione, 43% fluopyram/trifloxystrobin, and 10% polyoxin—were purchased from the Agricultural Resources Platform of IPP-CAAS (Appendix A).

### 2.2. Determination of the Effect of Wuyiencin on Botrytis cinerea Pathogenicity

#### 2.2.1. Colony Mycelial Growth and Conidial Germination

The inhibitory effect of wuyiencin on the colony and mycelial growth of *B. cinerea* was examined on PDA plates using the colony growth method [36]. In addition, the effect of wuyiencin on *B. cinerea* conidial germination was assessed; for specific details, please refer to Lv [30].

#### 2.2.2. Infection Pad Formation

Collected grape leaves were placed in 15 cm Petri dishes and sprayed with 0 (control), 20, 60, and 100 µg mL^−1^ of wuyiencin, respectively. Each concentration treatment was replicated three times. The Petri dishes were then placed in a growth chamber (25 °C, RH > 90%, 11,000 lx, and 12 light h/day), and the leaves were inoculated with *B. cinerea* mycelial discs (5 mm in diameter) at 24 h after wuyiencin treatment [37].

Four days after inoculation, three leaf pieces (5 × 5 mm) were prepared from leaf areas colonized by fungal hyphae in different treatment groups. The leaf pieces were fixed with 2.5% glutaraldehyde and kept at 4 °C overnight. The pieces were then washed thrice with 0.1 mol L^−1^ phosphate buffer for 10 min each and dehydrated in an ethanol gradient series (20–100%, with an increment of 20%) and treated in ethanol: isoamyl acetate solutions (1:1 and 1:2) for 10 min each. They were then dried with a carbon dioxide flow dryer and coated with gold ions for 5 min using an ion sputter. Finally, the leaf pieces were observed using scanning electronic microscopy (SEM), and the infection pads in different treatment groups were photographed to examine the morphological changes in *B. cinerea* infection pads under treatment with different concentrations of wuyiencin.

#### 2.2.3. Oxalic Acid Production

Twelve flasks containing 100 mL of potato dextrose broth, each with 10 mycelial discs (5 mm in diameter) per flask were placed on a rotary shaker at 200 rpm and 25 °C for 36 h. Then, appropriate quantities of wuyiencin were added to the flasks to make solutions containing 0 (as control), 20, 50, and 100 µg mL^−1^ of wuyiencin, respectively; three replicate flasks were established per concentration. After further shake-culturing for 0, 1, 3, 5, and 7 days, the mixture solutions were filtered through bacterial filters, and the filtrates were centrifuged at 12,000× *g* rpm for 10 min. The oxalic acid content in the supernatant was measured with a spectrophotometer (OD_510nm_). The oxalic acid content was calculated from the OD values using the method described by Duan [38] and compared between different treatment groups.

#### 2.2.4. Pathogenicity Gene Expression

Healthy grape leaves collected from greenhouse seedlings were placed on absorbent cotton in Petri dishes (15 mm in diameter) and sprayed with wuyiencin solution (65 µg mL^−1^) or with sterile water as the control. Both the wuyiencin treatment and the control were replicated three times. The leaves were then placed in a growth incubator (25 °C, RH > 90%, and 12 light h/day at 11,000 lx) and inoculated 24 h later by placing mycelium discs at appropriate locations on the leaf surface. At 6, 24, 48, 96, and 144 h after inoculation, infected/diseased foliar tissues (0.8 g/sample) were prepared aseptically from the leaves and used for RNA extraction.

Four important pathogenicity genes (Appendix A) of *B. cinerea* were selected for expression analysis; their gene sequences were downloaded from NCBI GenBank, and primers were designed (Appendix A) using Primer 5.0. The primers were synthesized by Beijing Invitrogen Company Ltd. (Invitrogen, Beijing, China) Total RNA was extracted from the treated and control leaf tissues using the total RNA extraction reagent (OMEGA, Bienne, Switzerland) and digested by DNase I (Promega, Madison, WI, USA) following the manufacturer’s instructions. cDNA was synthesized by reverse transcription using the PrimeScript TMRT Reagent Kit with gDNA Eraser (Takara, Otsu, Japan). Real-time qPCR was performed with the ABI 7500 qPCR amplifier (Applied Biosystems, Foster, CA, USA). The actin gene of *B. cinerea* (*BcAct*) was used as the internal reference, and the relative expression levels of the tested genes were calculated using the 2^−ΔΔCt^ method [39].

#### 2.2.5. Leaf and Fruit Disease Suppression

One healthy grape leaf and six grapefruits (just matured) were collected and placed on absorbent cotton in Petri dishes (15 mm in diameter) and treated with the corresponding wuyiencin solution (0, 50, 80, 100, or 120 µg mL^−1^); each concentration treatment was performed in triplicate. The Petri plates were then placed in a growth chamber (25 °C, RH > 90% and 12 light h/day at 11,000 lx), and the mycelium discs were placed on the leaf surface for inoculation 24 h later, while the fruits were inoculated with 10 µL of conidial suspension (3.3 × 10^6^ spores/mL) at three points around the navel. The lesions of leaves and fruits were measured after inoculation for four days and seven days, respectively. Gray mold disease suppression by wuyiencin was determined using the following formula:

Disease suppression (%) = (lesion diameter of control leaves − lesion diameter of wuyiencin-treated leaves)/lesion diameter of control leaves × 100.

Disease suppression (%) = (disease incidence on control fruits − disease incidence on wuyiencin-treated fruits)/disease incidence on control fruits × 100.

### 2.3. Assessment of the Plant Defense Response Induced by Wuyiencin

#### 2.3.1. Changes in Leaf Cellular Organelles

The protocol used here was similar to that used to assess infection pad formation. Healthy leaves were sprayed with wuyiencin at concentrations of 50, 80, and 100 µg mL^−1^, respectively. The leaves were stored at 25 °C for one day, then inoculated with *B. cinerea*. After 4 days, 3 × 3 mm leaf pieces were prepared from leaf areas colonized with fungal hyphae in different treatment groups. Leaf samples were prepared for transmission electron microscopy (TEM), and leaf tissue sections were photographed to observe morphological changes in cellular organelles (conidia formed in tissues, cell membranes, chloroplasts, and starch granules) caused by *B. cinerea* infection and treatment with different wuyiencin concentrations [40].

#### 2.3.2. Callose Deposition

Grape leaves were harvested from healthy plants (rose-scent), placed on wet degreased cotton in Petri dishes (9 cm in diameter), and sprayed with wuyiencin solution (50 µg mL^−1^). They were incubated for 24 h in a growth chamber (25 °C, RH > 90%, 11,000 lx, and 12 light h/day), then inoculated with *B. cinerea* suspension (3.3 × 10^6^ spores/mL), and further incubated for 24 h. Leaf sections in the control group were not inoculated and were sprayed with sterile water; those in the experimental group were inoculated and sprayed with sterile water. Each treatment was performed in triplicate. Leaf sections (10 × 10 mm) prepared from leaves in different treatment groups were decolorized with PBS buffer (phenol:glycerol:lactic acid:water:ethanol = 1:1:1:8 by volume), stained with K_2_HPO_4_ solution containing 0.01% aniline blue (pH 9.5), and viewed and photographed using a BX61 bio-fluorescent microscope (Olympus, Tokyo, Japan).

#### 2.3.3. Reactive Oxygen Species Accumulation

Reactive oxygen species (ROS) accumulation in plant leaves was determined using the 3,3-diaminobenzidine (DAB) staining method [41,42]. Four treatment groups were established: sterile water (control), *B. cinerea* inoculation, 50 µg mL^−1^ wuyiencin + *B. cinerea* inoculation, and 100 µg mL^−1^ wuyiencin + *B. cinerea* inoculation. Each treatment was performed in triplicate, and the treated leaves were harvested at 12, 24, 48, and 72 h, respectively.

Leaf pieces (10 × 10 mm) from each treatment group were completely immersed in DAB solution (1 mg mL^−1^) and warmed in a water bath at 35 °C for 4 h. The stained leaves were removed from DAB, immersed in 75% alcohol, and heated at 70 °C for 15 min; this process was repeated three times to ensure that chlorophyll was completely removed from the leaf tissues. Finally, the leaf pieces were observed under an inverted fluorescence microscope, and photographs were taken under visible light. Reddish-brown spots on the images indicated accumulated ROS in leaf tissues, and the level of ROS accumulation in different treatment groups was assessed accordingly.

### 2.4. Selection of Effective Mixed Preparations of Wuyiencin and Fungicides for Gray Mold Control

#### 2.4.1. Screening Effective Chemical Fungicides

The inhibition of *B. cinerea* by four common chemical fungicides (40% pyrimethanil, 50% iprodione, 43% fluopyram/trifloxystrobin, and 10% polyoxin) was assessed using the colony growth method [36]; wuyiencin was used as the control. PDA plates (9 cm in diameter) containing different concentrations (0, 20, 40, 60, 80, 100, and 120 µg mL^−1^) of each fungicide were prepared, and three replicates were used per concentration. Mycelial discs (6 mm in diameter) of *B. cinerea* were prepared, and one disc was placed at the center of each PDA plate. All Petri dishes were incubated at 25 °C in the dark for 3 days. The fungal colonies were photographed; the colony diameter was recorded, and the colony growth inhibition rate was calculated. To further obtain the median effective concentration (EC_50_) of chemicals fungicides, the tested concentrations were set at corresponding limits (pyrimethanil: 1.5–24 mg L^−1^; iprodione: 1–16 mg L^−1^; fluopyram/trifloxystrobin: 0.25–4 mg L^−1^, polyoxin: 30–480 mg L^−1^) according to the pre-experiment. The virulence function was established using the linear regression analysis, and the median effective concentration (EC_50_) was computed from the linear function for each fungicide. Finally, highly effective fungicides (with low EC_50_ values) were selected as candidates for further tests using mixed preparations of chemical fungicides with wuyiencin.

#### 2.4.2. Determination of the Optimal Ratio of Candidate Fungicides in Mixed Preparations

The method described by Horsfall [43] was employed. Wuyiencin was mixed with each candidate fungicide in different proportions (*v*:*v* ratios = 0:10, 1:9, 2:8, 3:7, 4:6, 5:5, 6:4, 7:3, 8:2, 9:1, and 10:0). PDA plates containing the mixtures were prepared; no fungicide was added in the control PDA plates. Each treatment was performed in triplicate. All plates were inoculated by placing a *B. cinerea* mycelial disc (6 mm in diameter) at the center of each plate and incubated at 25 °C in the dark for 3 days. The fungal colony diameter was measured, and the colony growth inhibition rate (%) was calculated for each treatment group. The optimal ratio for the mixed preparation was determined based on the estimated colony growth inhibition rate and the toxicity ratio, which were calculated with the following formulae:

Colony growth inhibition rate (%) = (% inhibition by wuyiencin at its EC_50_ × % of wuyiencin) + (% inhibition by chemical fungicide at its EC_50_ × % of chemical fungicide)

Toxicity ratio of the preparation = % inhibition by the preparation/estimated colony growth inhibition effect (%).

A mixed preparation of wuyiencin and chemical fungicide with a toxicity ratio of >1.0, 1.0, or <1.0 were considered to have a positive (synergistic) effect, no (additive) effect, or negative (mutual inhibitive) effect, respectively.

#### 2.4.3. Selection of the Mixed Preparation with the Highest Synergism

Mixed preparations of fungicides with synergistic effects (as indicated by the toxicity ratio test) were further tested to identify the preparation with the strongest synergistic effect; plate preparation and pathogen culturing were performed using the methods described above. After measuring the colony diameter, the colony growth inhibition rate, virulence regression function, and EC_50_ for each mixed preparation were computed. The interaction types of fungicides in different mixed preparations were determined according to the co-toxicity coefficient (CTC) using the method described by Sun and Johnson [44]: CTC < 80, 80 ≤ CTC ≤ 120, and CTC > 120 indicated an antagonistic, additive, and synergistic interaction, respectively. The fungicidal preparation with the highest CTC and lowest EC_50_ was identified as the optimal mixed preparation.

#### 2.4.4. Evaluation of *Botrytis cinerea* Colony Growth Inhibition by the Optimal Mixed Preparation

The treatments in this experiment included fungicide A (EC_50_), fungicide B (EC_50_), the selected best-mixed preparation with the highest ratio of fungicides A and B, and the control (no fungicide). Each treatment was performed in triplicate. PDA plates were prepared for different treatment groups, and a mycelial disc (5 mm in diameter) was placed at the center of each PDA plate. The plates were incubated at 25 °C in the dark for 3 days. Then, the colony diameter was measured, and colony growth inhibition rates were calculated and compared among treatment groups.

#### 2.4.5. Evaluation of Gray Mold Disease Suppression by the Optimal Mixed Preparation on Leaves and Fruits

The treatments in this experiment included fungicide A (EC_50_), fungicide B (EC_50_), the optimal mixed preparation with the highest ratio of fungicides A and B, and control (no fungicide). Each treatment was performed in triplicate.

### 2.5. Effect of Wuyiencin on Fungicide Resistance in Botrytis cinerea and Its Mechanism

#### 2.5.1. Classification of *Botrytis cinerea* Strains Based on Fungicide Resistance Levels

The mycelial growth rate method was used to determine the fungicide resistance of *B. cinerea* strains. Fungal cakes (6 mm in diameter) were punched out at the edge of colonies formed on PDA plates (after culturing at 25 °C in the dark for 3 d). The doses of pyrimethanil used were 0, 5, 25, 50, 100, and 125 mg L^−1^. The fungal cakes were transferred to the medium plates containing pyrimethanil, and the plates were incubated in the dark at 25 °C for 3 d. Each treatment was repeated five times, and the experiment was repeated three times. The radial growth of colonies was measured to determine the inhibition rate of mycelial growth. The sensitivity of different strains of gray mold to pyrimethanil was determined by performing a linear regression analysis between the probability of mycelial growth inhibition and the log-transformed concentration of pyrimethanil.

Growth inhibition (%) = [(control colony diameter − treatment colony diameter)/control colony diameter] × 100.

Pyrimethanil resistance was determined based on *B. cinerea* baseline sensitivity to pyrimethanil (0.091 mg L^−1^):

Resistance level = strain EC_50_/sensitivity baseline.

A resistance level ≤ 10 indicated pyrimethanil sensitivity; 10 < resistance level ≤ 50 and 50 < resistance level ≤ 100 indicated low and moderate resistance to pyrimethanil, respectively; and a resistance level > 100 indicated high resistance to pyrimethanil.

#### 2.5.2. Determination of the Retardation Effect of Wuyiencin on the Resistance of *Botrytis cinerea* Strains

Pyrimethanil (alone and in combination with wuyiencin) was used to eliminate the resistance of *B. cinerea* strain B05.10 and to determine whether wuyiencin could retard the development of pyrimethanil resistance in the B05.10 strain. Tests were conducted using a concentration gradient (8, 10, 16, 20, and 24 µg mL^−1^) of pyrimethanil EC_50_ and pyrimethanil EC_50_+wuyimycin EC_50_ and PDA medium, after which, generations 1, 8, and 10 of the different strains were selected for the inhibition assay. Subsequently, the virulence regression equation was fitted using SPSS analysis software to generate the virulence regression curve, and the EC_50_ value was derived for the calculation of resistance multiplicity.

#### 2.5.3. Mechanism of the Retardation of Resistance in *Botrytis cinerea* by Wuyiencin

The effect of wuyiencin on the activity of cell wall-degrading enzymes, such as β-glucosidase, pectinase, and cellulase, was determined using a commercial kit (purchased from Beijing Box Company Biotech Co., Ltd. (Box, Beijing, China). The mycelial growth rate method was used to determine the effect of pyrimethanil on the activity of cytosolic hydrolase. Pyrimethanil-resistant strains were cultured for 5 d. Using a sterile punch with a diameter of 1 cm, a few fungal cakes were punched at the edge of the colony and set aside. The combination of wuyiencin + pyrimethanil was added to 100 mL PDB to obtain the final concentrations of 0, 8, 10, 16, 20, and 24 µg mL^−1^ of fungicide-containing medium. PDB without wuyiencin was used as a blank control, and each treatment was repeated three times. After that, the medium was incubated at 25 °C with shaking at 160 rpm for 5 d and removed for use. The mycelium blocks were removed from the PDB medium, placed in liquid nitrogen, completely ground according to a certain ratio, and placed on ice. The ground mycelium was transferred to a 2 mL centrifuge tube. Then, the extract was added according to the instructions of the enzyme activity assay kit and homogenized in an ice bath. The homogenate was centrifuged at 10,000× *g* at 4 °C for 10 min, and the supernatant was placed on ice for further analysis.

#### 2.5.4. Determination of the Effect of Wuyiencin on Expression of the Methionine Biosynthesis Gene

The sequence of the mutated gene was queried using NCBI, and the effect of pyrimethanil on the transcriptional levels of the mutated gene in the resistant strain was determined using the mycelial growth rate method. Pyrimethanil-resistant strains were cultured for 5 d. Using a sterile punch with a diameter of 6 mm, a few fungal cakes were punched at the edge of the colony and set aside. The prepared fermentation broth containing wuyiencin was placed on an ultra-clean bench and filtered through a 0.22 µm bacterial filter (Millex-GP) before use. The filtered fermentation broth was mixed with 100 mL of sterilized PDA to obtain final concentrations of 0, 20, 40, 50, and 60 µg mL^−1^ of drug-containing medium. PDA medium without wuyiencin was used as a blank control, and each treatment was repeated three times. The media were shaken well, poured into a sterile medium, and allowed to solidify. Then, a fungal cake was placed at the center of the medium using a sterile inoculation needle, face down, and incubated until 4 d for observation.

### 2.6. Statistical Analysis of the Data

According to the statistical results of the experiment, the data were transformed using arcsine transformation. The test data were calculated and plotted using Excel 2020 software. One-way ANOVA in SPSS 22.0 software was employed to assess the differences among the mean values of each treatment. Duncan’s multiple comparison test was conducted to determine the significance of the observed differences. The virulence regression curve was synthesized and analyzed through regression probability analysis.

## 3. Results

### 3.1. Inhibitory Effect of Wuyiencin on Botrytis Cinerea Pathogenicity

#### 3.1.1. Colony and Mycelial Growth

The application of wuyiencin had a significant inhibitory effect on the colony and mycelial growth of *B. cinerea*. The increase in wuyiencin concentration strengthened the inhibitory effect. The colony diameter in the 60 and 120 µg mL^−1^ wuyiencin treatment groups (29.9 ± 2.4 mm and 3.7 ± 1.1 mm, respectively) was lower than that in the control group (65.0 ± 2.6 mm), representing colony growth inhibition rates of 54.0% and 94.3% in the two treatment groups, respectively (Appendix A).

Using the colony growth inhibition rates of different concentrations of wuyiencin (Appendix A), the toxicity regression line of wuyiencin was produced (Appendix A), and the computed linear function was y = 0.6155x + 16.7910 (*R*^2^ = 0.8534, *p* < 0.01), where x and y stand for wuyiencin concentration and colony growth inhibition rate, respectively. The median and absolute effective concentrations (EC_50_ and EC_100_) calculated from the function were 53.954 and 135.189 µg mL^−1^, respectively, indicating that wuyiencin was highly effective in suppressing the colony growth of *B. cinerea*.

Microscopic observations indicated that wuyiencin treatment markedly altered the mycelial morphology of *B. cinerea*. In the control treatment, the hyphae showed abundant growth and were comparatively straight, smooth, hyaline, and branching (mostly at acute angles), with a uniform cell size and cell wall thickness. In comparison, the mycelia of wuyiencin-treated *B. cinerea* showed abnormal growth, and the abnormality became more severe and evident as wuyiencin concentration increased from 20 µg mL^−1^ to 100 µg mL^−1^ (Appendix A). Specifically, in the 100 µg mL^−1^ treatment, the hyphae were severely curved, shrunken, and branching mostly at right angles; their color deepened or darkened, and shorter cells and thicker cell walls were observed.

#### 3.1.2. Conidial Germination

In the control treatment without wuyiencin, 97.2% of *B. cinerea* conidia germinated, and hyphae formed densely from the germinated conidia after incubation at 22 °C under humid and dark conditions for 24 h. Wuyiencin treatment strongly suppressed spore germination. With increasing wuyiencin concentration from 20 µg mL^−1^ to 120 µg mL^−1^, the conidial germination rate decreased from 61.8% to 2.1% (Appendix A). In the higher concentration treatment groups (≥60 µg mL^−1^), only a few conidia germinated to form short and malformed germ tubes, which could not develop further to produce normal hyphae (Appendix A).

#### 3.1.3. Infection Pad Formation

The infection pad formation of *B. cinerea* was affected by wuyiencin application (Appendix A). When no wuyiencin was applied after inoculation (control treatment), the pathogen produced more and denser infection pads on the leaf surface, with abundant and uniform hyphae. After inoculation, some hyphal tips enlarged and thickened and branched three or four times to form mature claw-shaped infection pads, which produced infection pegs that penetrated the epidermis and entered the leaf tissue. However, when the leaves were inoculated and treated with wuyiencin, the infection pad formation was inhibited, with fewer and sparser infection pads observed on the leaf surface. Compared with those in the control treatment, the infection pads in the wuyiencin treatment groups (especially 100 µg mL^−1^) were weaker and showed fewer hyphae; the hyphae were short, slender, shriveled, and cracked at the surface; their tips had not enlarged to form infection pegs, and they could barely penetrate the epidermis to enter the leaf tissue.

#### 3.1.4. Oxalic Acid Production

Oxalic acid secreted by *B. cinerea* increases the acidity of the host surface, which facilitates the invasion and colonization of the host by *B. cinerea*. Thus, oxalic acid plays an indispensable role in the infection process of *B. cinerea* [45]. The findings (Appendix A) show that the concentration of oxalic acid synthesized in leaf tissues (in the control and all wuyiencin concentration treatments) initially increased, reaching the highest concentration at 3 days after *B. cinerea* inoculation, and gradually decreased from 3 to 7 days after inoculation. However, there was a significant difference in oxalic acid content between the control and wuyiencin treatment groups (*p* < 0.05) from 3 to 7 days after inoculation. In particular, on day 3 after inoculation, the oxalic acid concentration in the control reached 43 µg mL^−1^, whereas the concentrations in the wuyiencin treatment groups ranged from 25 to 29 µg mL^−1^.

#### 3.1.5. Expression of Pathogenicity Genes

The expression of four pathogenicity genes of *B. cinerea*—*Bcpls*, *Bcmp1*, *Bcpg1*, and *BcnoxA*—in grape leaf tissues was assessed to examine the effect of wuyiencin. Experimental results (Figure 1) indicated that all four genes showed similar expression patterns after *B. cinerea* inoculation. In the control treatment (without wuyiencin application), the gene expression increased from low to rather high levels as time after inoculation progressed from 6 h to 144 h. However, wuyiencin application markedly affected the expression of these genes; their expression levels in the wuyiencin treatment group decreased as time after inoculation progressed from 6 h to 144 h. Compared with those in the control group, the gene expression levels in the wuyiencin treatment group were significantly lower (*p* < 0.01) starting from 24 h after *B. cinerea* inoculation.

#### 3.1.6. Disease Suppression on Leaves and Fruits

When no wuyiencin was applied (control), gray mold disease developed rapidly on inoculated grape leaves; the disease lesions were visible within 24 h and showed a mean diameter of 5.9 cm at 4 days after inoculation (Appendix A). Wuyiencin treatment severely suppressed disease development. The disease suppression rate increased with the increase in wuyiencin concentration and reached 88.1% in the 120 µg mL^−1^ wuyiencin treatment group (Appendix A). Regarding disease symptoms (Figure 2), the disease spots were irregular, brown, and large in the control group and low wuyiencin concentration (50 µg mL^−1^) treatment group; most parts of the leaves turned to be brown colored and covered with gray mold (mycelia of *B. cinerea*). However, in groups treated with higher wuyiencin concentrations (≥80 µg mL^−1^), the spots were smaller, and the areas with brown and dead tissue were limited. Notably, in the 120 µg mL^−1^ wuyiencin treatment group, the leaves remained almost completely healthy and the spots were circular, slightly brown, and restricted to the points of inoculation.

When measuring the control effect of wuyiencin on grapefruit, the results showed that disease incidence on fruits treated with different concentrations of wuyiencin was significantly lower than that in the control group (*p* < 0.01). The fruit gray mold disease was almost completely suppressed (96.6%) when 120 µg mL^−1^ wuyiencin was applied. Regarding disease symptoms (Figure 2), in the control group (with no wuyiencin), the fruits were almost completely rotten, and their surface was covered with abundant gray mold (fungal mycelia). The symptoms were substantially alleviated when the fruits were treated with different concentrations of wuyiencin. In the high-concentration treatment groups (100 and 120 µg mL^−1^), the disease spots were markedly smaller, and gray mold mycelia were almost undetectable.

### 3.2. Assessment of the Plant Defense Response Induced by Wuyiencin and Its Control Effect

#### 3.2.1. Change in Leaf Cellular Organelles

Wuyiencin application at different concentrations affected *B. cinerea* infection and the morphology of some cellular organelles in grape leaf tissues (Figure 3). After grape leaves were inoculated with *B. cinerea* mycelial discs on the abaxial surface without wuyiencin application (control), the fungal conidia were detected in the palisade tissues of the adaxial surface (Figure 3a,c), indicating that *B. cinerea* successfully infected the leaf tissue and became established in the leaf tissues; it produced normal conidia with intact cellular organelles (Figure 3d). The leaf cells showed dissolved and collapsed cell walls (Figure 3b), malformed or collapsed chloroplasts (Figure 3b), and barely visible starch granules.

Wuyiencin treatment (50, 80, and 100 µg mL^−1^) markedly impeded pathogen infection and protected the host tissues. When wuyiencin was applied before *B. cinerea* inoculation, a few conidia were observed only in the spongy tissues near the inoculated leaf surface, but none were found inside the palisade and mesophyll tissues (Figure 3i,j,m,n), indicating that the pathogen infection was severely hindered. The conidial cells were seriously malformed, and their cellular organelles mostly collapsed, resulting in vacuolar conidial cells (Figure 3k,l,n,o). In the leaf tissue, most cell walls were visible with good structure (Figure 3b,k,o). The chloroplasts were in good condition, and starch granules were clearly visible in leaf cells (Figure 3j,k,p).

#### 3.2.2. Callose Deposition

Callose in plant tissues acts as a physical barrier to infection by pathogens [46]. The results (Appendix A) showed that little callose deposition (florescent blue–white dots) was detected in the leaf tissue without inoculation and wuyiencin treatment. There was some callose deposition observed in the leaf tissues treated only with wuyiencin (50 µg mL^−1^). After leaf tissue was sprayed with wuyiencin and inoculated with *B. cinerea*; however, callose deposition in the tissues markedly increased. This indicated that wuyiencin treatment induced callose accumulation in inoculated grape leaf tissues, which enhanced the physical defense of grape vines and thus effectively inhibited the infection by *B. cinerea*.

#### 3.2.3. Reactive Oxygen Species Accumulation

ROS (H_2_O_2_) accumulation is a common response to fungal infections in host plant tissues [47]. The results (Figure 4) indicated that no H_2_O_2_ was detected in grape leaf tissues in the control group, and only trace amounts of H_2_O_2_ were observed in *B. cinerea*-inoculated leaves at 12 h to 72 h after treatment. However, under wuyiencin application at both concentrations (50 and 100 µg mL^−1^) following *B. cinerea* inoculation, the amount of H_2_O_2_ detected in the treated leaf tissues was markedly greater than that in the leaf tissues without wuyiencin treatment. Moreover, the H_2_O_2_ accumulation in the leaf tissue increased with increasing wuyiencin concentration and increasing time after treatment.

### 3.3. Compound Preparations of Wuyiencin and Chemical Fungicides against Grape Gray Mold

#### 3.3.1. Effective Chemical Fungicides

The results (Appendix A) showed that wuyiencin applied at different concentrations had a strong inhibitory effect on *B. cinerea* colony growth. Of the four fungicides tested, iprodione, pyrimethanil, and fluopyram/trifloxystrobin strongly suppressed *B. cinerea* growth. The fungal growth was almost completely inhibited when these fungicides were applied at higher concentrations (≥80 µg mL^−1^). However, polyoxin did not show a significant inhibitory effect at any concentration, and the colony size and mycelial abundance were similar to those observed in the control group (no fungicide).

The virulence regression functions were obtained, and EC_50_ values of the tested fungicides were computed (Table 1). All regression functions were highly significant (*p* < 0.01), with high determination coefficients (r^2^). A similar inhibitory effect on *B. cinerea* growth was illustrated by the EC_50_ values: the ineffective fungicide polyoxin showed an extremely high EC_50_ value (290.59 mg L^−1^), whereas the other three highly effective chemical fungicides showed low EC_50_ values (1.207–51.23 µg mL^−1^).

#### 3.3.2. Synergistic Interaction of Wuyiencin with Candidate Fungicides

According to the results of the above experiments, iprodione, pyrimethanil, and fluopyram/trifloxystrobin (fungicides with low EC_50_) were selected as candidate fungicides to be mixed, respectively, with wuyiencin in different proportions. The toxicity ratios and interaction type of fungicides in the mixed preparations are given in Table 2. Mixed preparations of wuyiencin with iprodione in different proportions showed antagonistic or additive interactions but no synergism. Mixed preparations of wuyiencin and pyrimethanil in ratios 6:4, 7:3, and 9:1 showed synergistic interactions, but their mixed preparations in other ratios showed additive or antagonistic interactions. Mixed preparations of wuyiencin with fluopyram/trifloxystrobin in ratios of 5:5, 6:4, 7:3, 8:2, and 9:1 showed synergistic interactions, and those in ratios of 1:9, 2:6. 3:7, and 4:6 showed additive interactions.

#### 3.3.3. Evaluating the Synergistic Effect of Mixed Preparations

Based on the results above, the three synergistic preparations of wuyiencin + pyrimethanil (6:4, 7:3, and 9:1) and three preparations of wuyiencin + fluopyram/trifloxystrobin (6:4, 8:2, and 9:1) were selected to further evaluate their synergistic effects. The testing results (Table 3) showed that the mixed preparations of wuyiencin + pyrimethanil in the ratios 7:3 and 9:1 showed the lowest EC_50_ (5.153 and 8.642 µg mL^−1^) and highest co-toxicity coefficient (CTC; 391.4 and 391.7), respectively. The mixed preparation of wuyiencin + fluopyram/trifloxystrobin also resulted in very low EC_50_ (4.845 µg mL^−1^) and high CTC (205.5). The EC_50_ of the other mixed preparations was high and their CTC was low considerably. To compare these results in an integrated manner, it was clear that wuyiencin + pyrimethanil with a ratio of 7:3 was the best one among all the mixed preparations tested.

#### 3.3.4. Inhibitory Effect of the Selected Mixed Preparation on *Botrytis cinerea* Colony Growth

The results of the above experiment showed that wuyiencin + pyrimethanil in the ratio 7:3 was the most effective among all the mixed preparations tested; therefore, this preparation was further examined for its inhibitory effect on *B. cinerea* colony growth. The results (Appendix A) showed that the inhibitory effect of the two fungicides and the selected compound preparation on *B. cinerea* was significantly higher than that in the control treatment (*p* < 0.01). Nevertheless, the compound preparation was the most effective in inhibiting *B. cinerea* colony growth, with an inhibition rate of 79.81% (Appendix A).

#### 3.3.5. Inhibitory Effect of the Optimal Mixed Preparation on *Botrytis cinerea* in Grape Leaves and Fruit

The selected optimal mixed preparation was tested to examine its inhibitory effect on gray mold development on grape leaves and fruits. The leaf and fruit experiments showed similar trends of gray mold suppression. In grape leaves (Table 4), both wuyiencin and pyrimethanil significantly suppressed the disease (*p* < 0.01) in comparison with the no fungicide control. However, the disease suppression rate of the mixed preparation was 51.05%, which was significantly higher than that of wuyiencin (47.56%) and pyrimethanil (29.37%). Similar results were obtained in fruits (Table 4), and the disease suppression effect of the mixture on fruits (66.06%) was even more prominent than that on leaves (Figure 2). The lesions on the leaves and fruits were large in the control group (Figure 2F) but markedly smaller or almost undetectable in the mixed preparation treatment group (Figure 2I). These results indicate that mixing the two fungicides is effective and necessary for disease suppression.

### 3.4. Wuyiencin Delayed Fungicide Resistance Development in Botrytis cinerea

#### 3.4.1. Classification of Different *Botrytis cinerea* Strains Based on Fungicide Resistance

Determination of fungicide sensitivity of six strains of *B. cinerea* (Table 5) showed that strain 503 was sensitive to pyrimethanil, whereas strains 57, 59, 61, 65, and 514 were resistant strains, with a high level of resistance to pyrimethanil (Table 6, Figure 5).

#### 3.4.2. Retardation of Resistance Development in *Botrytis cinerea* Strains by Wuyiencin

Delay in resistance development was measured by in vitro induction of highly resistant strains (Table 7). Using a 10th generation assay, the resistant strains were tested in culture using two fungicidal agents: pyrimethanil alone and wuyiencin plus pyrimethanil combination. It was found that the resistant strains induced by pyrimethanil were resistant at the 10th generation, and the peak resistance was 9.18-fold higher in the 10th generation. In contrast, the resistance multiplicity was only 1.27 in the resistant strains induced by the fungicide mixture. The low and steady increase in the resistance of the resistant strains induced by the combination confirmed that the addition of wuyiencin was effective in delaying resistance development.

### 3.5. Mechanism of Retardation of Resistance Development in Botrytis cinerea by Wuyiencin

#### 3.5.1. Effect of wuyiencin on Cell Wall-Degrading Enzyme Activity in Resistant *Botrytis cinerea*

Changes in β-glucosidase activity: Treatment with pyrimethanil alone and in combination with wuyiencin did not significantly affect the β-glucosidase activity of the highly pyrimethanil-resistant strain 57 (Figure 6); the β-glucosidase activity remained unchanged at approximately 302 U/g.

Changes in pectinase activity: Pyrimethanil treatment did not significantly affect the pectinase activity of the highly resistant strain 57; overall, it remained virtually unchanged (approximately 14 U/g) (Figure 7). When treated with the combination wuyiencin + pyrimethanil, the pectinase activity of the highly pyrimethanil-resistant strain 57 gradually increased at lower concentrations of wuyiencin + pyrimethanil, peaked (>38 U/g) at 10 µg mL^−1^ of wuyiencin + pyrimethanil, and then gradually decreased with the increasing concentration of wuyiencin + pyrimethanil. This finding indicates that pectinase activity was significantly inhibited by the combination of wuyiencin and pyrimethanil.

Changes in cellulase activity: The lowest cellulase activity of the highly pyrimethanil-resistant strain 57 after treatment with pyrimethanil alone was 15 U/g (Figure 8). In contrast, the highest cellulase activity observed was approximately 150 U/g after treatment with 8 µg mL^−1^ of wuyiencin + pyrimethanil, after which it decreased significantly to approximately 50 U/g. Overall, the combination of wuyiencin and pyrimethamine had a significant inhibitory effect on cellulase activity.

#### 3.5.2. Effect of Wuyiencin on the Expression Levels of Mutant Genes of *B. cinerea*

The expression levels of the methionine biosynthesis genes *CGS* and *CBL* of *B. cinerea* 57 (a highly pyrimethanil-resistant strain) were analyzed after treatment with wuyiencin (Figure 9). It was found that wuyiencin significantly inhibited the expression of these genes, and the inhibitory effect became stronger with increasing concentration of wuyiencin. The relative expression of *CGS* and *CBL* was downregulated by 4.38- and 2.85-fold, respectively, when the concentration of wuyiencin reached 60 µg mL^−1^.

## 4. Discussion and Conclusions

Presently, chemical fungicide application remains the main and most common method for controlling gray mold (*B. cinerea*), but it is widely known that chemical fungicides contaminate crop products and the environment [48,49]. Chemical fungicides inhibit *B. cinerea* by targeting the expression of specific single DNA sites [50]; therefore, genetic mutations leading to chemical fungicide resistance are easily induced in *B. cinerea* [51]. The application of biocontrol agents has been demonstrated to be a good alternative strategy for crop disease control, as they cause no significant contamination of crop products and the environment and decrease or avoid fungicide resistance. The biocontrol agent wuyiencin is a secondary metabolite produced by *S. albulus* var. *wuyiensis*, which has activity against various fungi and viruses [32]. The present findings showed that wuyiencin treatment strongly suppressed the growth and development of *B. cinerea* and significantly reduced the disease severity of grape gray mold.

Plant disease development is correlated with a number of complicated infection processes determined by the growth, sporulation, infection, and pathogenicity of a pathogen and responses of the host plant to the infection; most of these processes involve various pathological mechanisms regulated by pathogenicity genes [52,53,54]. Several recent studies have documented the mechanisms of inhibition of pathogen infection by biocontrol agents. Phytopathogens effectively infect their host plants usually by regulating the pathogenicity genes, increasing oxalic acid content, and producing some active extracellular enzymes and phytotoxins [45,55,56]. *Pseudomonas* 14D5 reduces Phytophthora blight on potato plants by inhibiting spore germination of the pathogen [57]. *Lactiplantibacillus plantarum* LPP703 suppresses the growth of fungi by degrading their membranes [58]. *Yarrowia lipolytica* can significantly control the occurrence of grape penicillium disease by inhibiting spore germination and bud tube length [59]. Different antibiotics such as validamycin, kasugamycin, and tetramycin produced by antagonists of plant pathogens have been exploited as biofungicides to control certain crop diseases [60]. Some antibiotics have a direct lethal effect on certain phytopathogens and induce resistance or immune responses to pathogen infections in the host plants. In the present study, wuyiencin application strongly inhibited mycelial growth, conidial germination, infection pad formation, and oxalic acid production of *B. cinerea*. Callose deposition and ROS accumulation were markedly increased, whereas the four pathogenicity genes (*Bcpls*, *Bcmp1*, *Bcpg1*, and *BcnoxA*) of *B. cinerea* were downregulated in the tissues of wuyiencin-treated grape leaves. These results suggest that wuyiencin induced the defense or resistance responses (callose and H_2_O_2_ accumulation) of the host plant and downregulated the expression of *B. cinerea* pathogenicity genes.

Owing to fungicide resistance and environmental contamination resulting from chemical control of plant diseases, mixed or compound use of a biocontrol agent with chemical fungicides has been proposed, which effectively reduces disease incidence, decreases chemical residue on agricultural products, and lessens environment contamination and fungicide resistance of the pathogen [18]. A previous study found that the combined use of a chemical fungicide and a biocontrol agent had a significantly higher inhibitory effect on chrysanthemum white rust (*Puccinia horiana* Henn.) compared to using only the fungicide, without affecting plant growth and development [61]. The combined use of a biocontrol agent with a chemical fungicide effectively controlled *Alternaria* leaf blight of ginseng and significantly reduced the dosage and frequency of chemical fungicide application in a cropping season [62]. In the present study, the compound preparations of wuyiencin and pyrimethanil, which were formulated in ratios 6:4, 7:3, and 9:1, produced significant synergistic effects on *B. cinerea* growth.

Based on this finding, an optimal compound preparation was formulated by mixing wuyiencin and pyrimethanil in a ratio of 7:3. This mixed preparation was significantly more effective (*p* < 0.05) than wuyiencin or pyrimethanil alone in controlling the gray mold disease on grape leaves and fruits. The use of wuyiencin significantly delayed resistance development in *B. cinerea* to pyrimethanil. Regarding the mechanism of its delay, it was observed that wuyiencin decreased the activity of the cell wall-degrading enzymes pectinase and cellulase and effectively reduced the expression of methionine biosynthesis genes. The present findings suggest that wuyiencin is a strong inhibitor of *B. cinerea* causing grape gray mold and an effective bio-fungicide for controlling the disease. Wuyiencin treatment strongly inhibited *B. cinerea* infection and, thus, protected the cellular organelles (cell walls and chloroplasts) in grape leaf tissues. Wuyiencin application also increased callose deposition and ROS accumulation in the treated plant tissues and downregulated *B. cinerea* pathogenicity genes (e.g., *Bcpls*, *Bcmp1*, *Bccpg*, and *BcnoxA*). Wuyiencin (EC_50_) combined with pyrimethanil (EC_50_) in the ratio of 7:3 was identified as the optimal mixed preparation for controlling grape gray mold. Wuyiencin had a strong inhibitory effect on highly pyrimethanil-resistant strains of gray mold and significantly delayed resistance development. Because of its unique mechanism of action, wuyiencin can be used in the field to control highly pyrimethanil-resistant strains of gray mold in combination with or as an effective alternative to pyrimethanil.

## Figures and Tables

**Figure 1 microorganisms-12-01383-f001:**
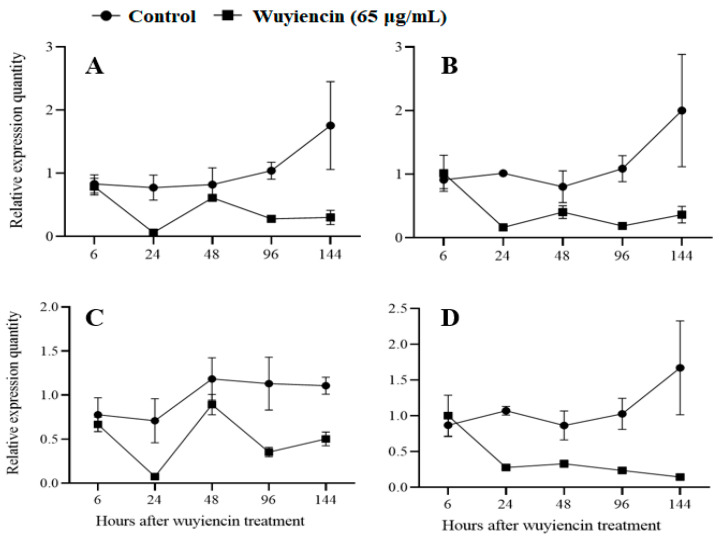
Effects of wuyiencin treatment on expression of four pathogenicity-related genes of *Botrytis cinerea* in grape leaf tissues. Note: Expression levels of the (**A**) transmembrane protein gene (*Bcpls*), (**B**) MAP kinase gene 1 (*Bmp1*), (**C**) NADPH oxidase gene A (*BcnoxA*), and (**D**) endopolygalacturonase gene 1 (*Bcpg1*) relative to that of the actin gene (*BcAct*; internal reference gene) at different time points (h) after wuyiencin (65 µg mL^−1^) and sterile water (control) sprays.

**Figure 2 microorganisms-12-01383-f002:**
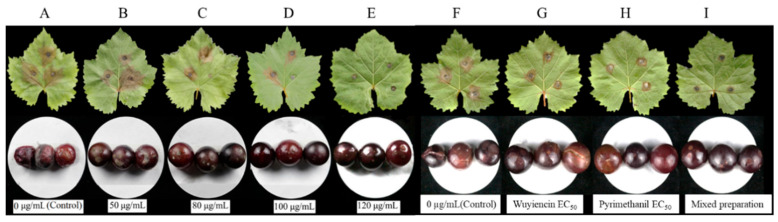
Effects of wuyiencin and a mixed fungicidal preparation on symptoms and severity of gray mold on grape leaves and fruits. Note: Wuyiencin was sprayed at different concentrations 24 h before leaves and fruits were inoculated with *Botrytis cinerea*. Images of leaves (upper) and fruits (lower) were taken, respectively, at 4 days and 7 days after *B. cinerea* inoculation (**A**–**E**). The letters (**F**–**I**) refer to control, wuyiencin at median effective concentration (EC_50_), pyrimethanil EC_50_, and the mixed preparation (wuyiencin EC_50_: pyrimethanil EC_50_ = 7:3).

**Figure 3 microorganisms-12-01383-f003:**
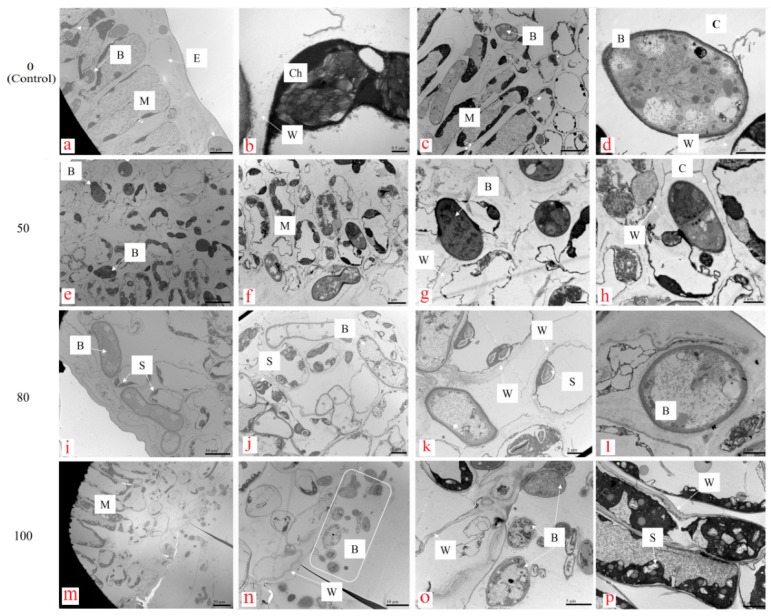
Effect of wuyiencin on the morphology of cellular organelles in grape leaf tissues inoculated with *Botrytis cinerea*. Note: Detached grape leaves were sprayed with 0 (control), 50, 80, and 100 µg mL^−1^ of wuyiencin and inoculated 24 h later with *B. cinerea* mycelial discs (5 × 5 mm). Scanning electron microscopy images were taken 4 days after *B. cinerea* inoculation. These images are: (**a**): leaf fenestrated tissue at 0 μg/mL wuyiencin; (**b**): chloroplasts at 0 μg/mL wuyiencin; (**c**): leaf spongy tissue and *B. cinerea* conidia at 0 μg/mL wuyiencin; (**d**): leaf endocarps at 0 μg/mL wuyiencin; (**e**–**h**): leaf spongy tissue at 50 μg/mL wuyiencin; (**i**–**l**): leaf spongy tissue, starch grains and *B. cinerea* conidia under 80 μg/mL wuyiencin; (**m**–**p**): leaf spongy tissue, fenestrated tissue, starch grains and *B. cinerea* conidia under 100 μg/mL wuyiencin.The capital letters refer to various cellular organelles: B, conidia produced by *B. cinerea*; M, mesophyll cell; Ch, chloroplast; E, epidermal cell; W, cell wall; CM, cell membrane; S, starch granule.

**Figure 4 microorganisms-12-01383-f004:**
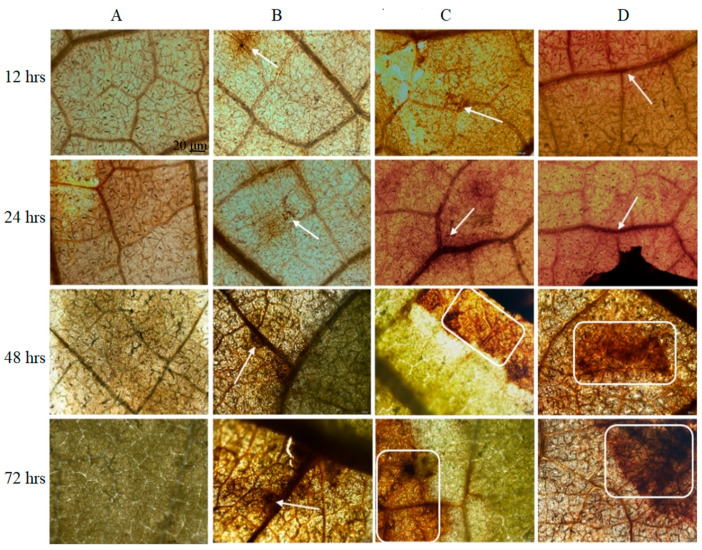
Effect of wuyiencin on reactive oxygen species (ROS; H_2_O_2_) accumulation in grape leaf tissues. Note: The reddish-brown spots (marked by white arrows or rectangles in the images) indicate accumulated ROS in leaf tissues. (**A**–**D**) represent different treatments (sterile water control, inoculated *B. cinerea*, *B. cinerea* inoculation + 50 µg mL^−1^ wuyiencin spray, and *B. cinerea* inoculation + 100 µg mL^−1^ wuyiencin spray), respectively. The time (h) after wuyiencin treatment is indicated on the left.

**Figure 5 microorganisms-12-01383-f005:**
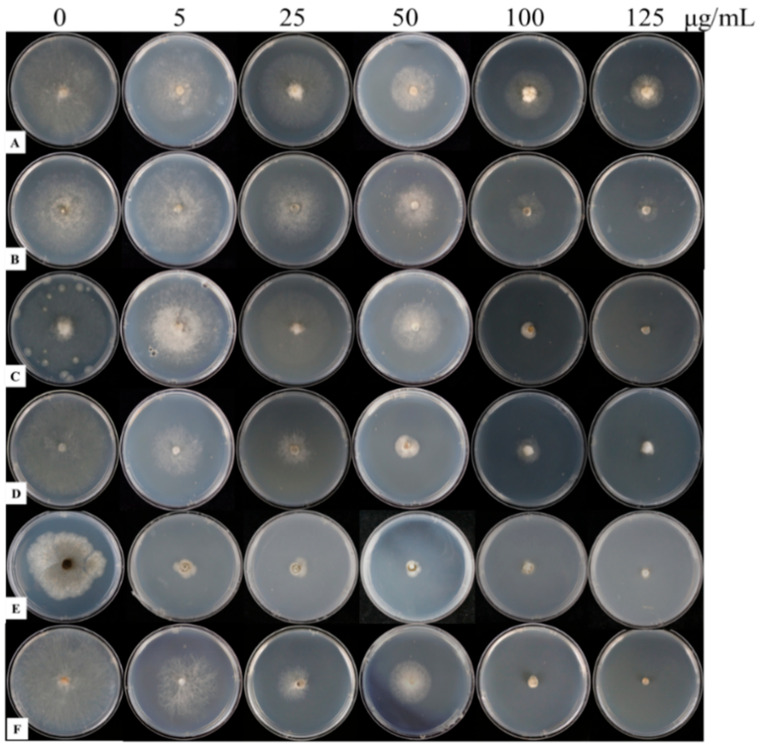
Inhibitory effect of different concentrations of pyrimethanil on colony growth of different *Botrytis cinerea* strains. Note: Images were taken 4 days after inoculation with mycelial discs (6 mm in diameter) of *B. cinerea*. The strains 57 (**A**), 59 (**B**), 61 (**C**), 65 (**D**), 503 (**E**), and 514 (**F**) were cultured on PDA plates; the concentrations of pyrimethanil were 0 (control), 5, 25, 50, 100, and 125 µg mL^−1^.

**Figure 6 microorganisms-12-01383-f006:**
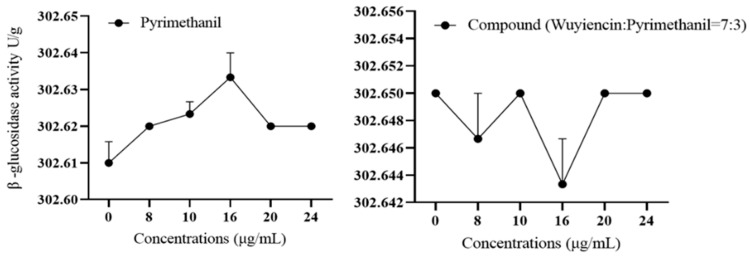
Change in β-glucosidase activity in *Botrytis cinerea* treated with pyrimethanil alone and wuyiencin plus pyrimethanil.

**Figure 7 microorganisms-12-01383-f007:**
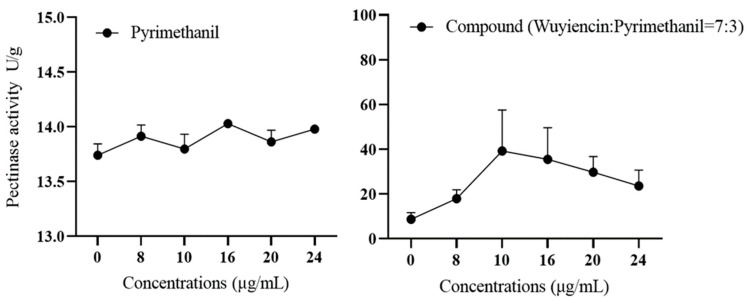
Change in pectinase activity in *Botrytis cinerea* treated with pyrimethanil alone and wuyiencin plus pyrimethanil.

**Figure 8 microorganisms-12-01383-f008:**
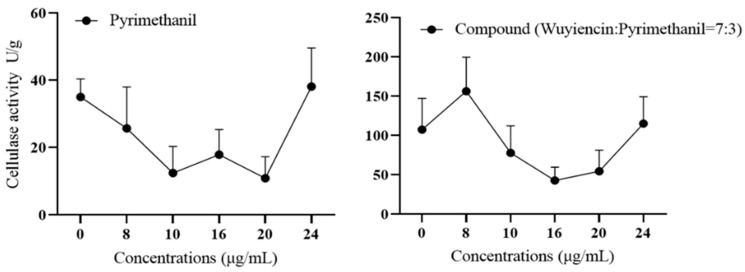
Change in cellulase activity in *Botrytis cinerea* treated with pyrimethanil alone and wuyiencin plus pyrimethanil.

**Figure 9 microorganisms-12-01383-f009:**
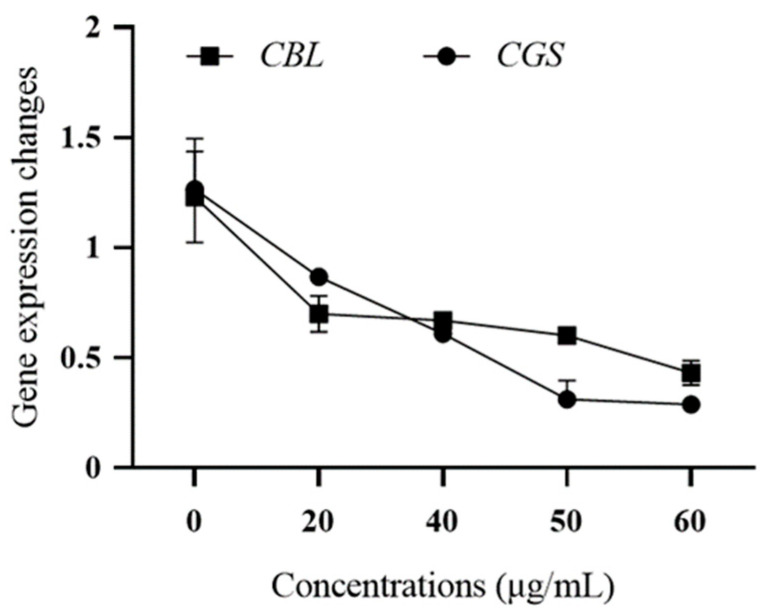
Effect of wuyiencin on the expression level of methionine biosynthesis gene of *Botrytis cinerea*.

**Table 1 microorganisms-12-01383-t001:** Virulence regression functions and EC_50_ values of the fungicides tested.

Fungicide	Virulence Function	r^2^	EC_50_ (mg L^−1^)	Range of EC_50_ (mg L^−1^)
Wuyiencin	y = 2.59x − 4.47 *	0.833	51.213	31.278–71.074
Pyrimethanil	y = 1.24x − 1.12 **	0.978	8.354	2.91–13.951
Polyoxin	y = 2.8x − 6.87 **	0.913	290.059	230.759–462.647
Iprodione	y = 1.99x − 0.97 **	1.000	4.724	0.216–9.728
Fluopyram/trifloxystrobin	y = 0.85x − 0.08 **	0.956	1.207	0.002–5.317

Note: x and y in the regression function present the log-transformed value of fungicide concentration and percentage of *B. cinerea* colony growth inhibition by the fungicide, respectively; one or two asterisks indicates that it is significant at *p* < 0.05 or *p* < 0.01; r^2^ is the determination coefficient of the function, and EC_50_ is the median effective concentration of a fungicide calculated from the function.

**Table 2 microorganisms-12-01383-t002:** Toxicity ratio (TR) values of wuyiencin mixed with three fungicides, respectively, against the colony growth of *B. cinerea* and interaction type of each pair of fungicides.

Wuyiencin +		10:0	1:9	2:8	3:7	4:6	5:5	6:4	7:3	8:2	9:1	0:10
Iprodione	TR	1.000	0.781	1.043	0.901	0.863	0.904	0.917	0.888	0.983	1.006	1.000
	Type	-	Ant.	Add.	Add.	Ant.	Add.	Add.	Ant.	Add.	Add.	-
Pyrimethanil	TR	1.000	0.777	1.011	1.045	1.045	1.046	1.085	1.085	1.014	1.091	1.000
	Type	-	Ant.	Add.	Add.	Add.	Add.	Syn.	Syn.	Add.	Syn.	-
Fluopyram· trifloxystrobin	TR	1.000	1.050	0.934	1.050	1.046	1.118	1.163	1.142	1.156	1.157	1.000
	Type	-	Add.	Add.	Add.	Add.	Syn.	Syn.	Syn.	Syn.	Syn.	-

Note: Ant: antagonistic; Add: additive; Syn: synergistic.

**Table 3 microorganisms-12-01383-t003:** Virulence regression functions and EC_50_ values of the mixed fungicide preparations.

Preparation of Wuyiencin (A) +	Ratio (A:B)	Virulence Function	r^2^	EC_50_ (μg mL^−1^)	CTC
Pyrimethanil (B)	10:0	y = 2.59x − 4.47 *	0.833	51.213	—
	6:4	y = 1.46x − 1.57 **	0.978	11.553	145.2
	7:3	y = 1.26x − 0.85 **	0.922	5.153	391.4
	9:1	y = 1.48x − 1.38 **	0.947	8.642	391.7
	0:10	y = 1.24x − 1.12 **	0.978	8.354	—
Fluopyram·trifloxystrobin (B)	10:0	y = 2.59x − 4.47 **	0.833	51.213	—
	6:4	y = 1.00x − 0.84 **	0.990	6.926	42.1
	8:2	y = 1.41x − 1.07 **	0.915	6.030	91.4
	9:1	y = 0.33x − 0.23 **	0.968	4.845	205.5
	0:10	y = 0.85x − 0.08 **	0.956	1.207	—

Note: In the functions, x and y present, respectively, logarithm-transformed value of fungicide concentration and probability of colony growth suppression of *B. cinerea* by the fungicide. The function with one or two asterisks indicates that it is significant at *p* < 0.05 or *p* < 0.01; r^2^ is the determination coefficient of the function and EC_50_ is the median effective concentration of a fungicide calculated from the function. The co-toxicity coefficient (CTC) was calculated following the steps of the Sun and Johnson method (1960).

**Table 4 microorganisms-12-01383-t004:** Inhibitory effect of the best-mixed preparation (wuyiencin + pyrimethanil) on *Botrytis cinerea*.

Fungicide	Effect of Leaves Control		Effect of Fruits Control	
	Lesion Diameter (cm)	Suppression Rate (%)	Disease Incidence (%)	Disease Suppression Rate (%)
Control	1.43 ± 0.34 a	0.0	76.40 ± 1.40 a	0.0
Wuyiencin EC_50_	0.75 ± 0.26 b	47.56	48.61 ± 1.39 c	36.37
PyrimethanilEC_50_	1.01 ± 0.28 ab	29.37	51.47 ± 1.40 b	32.63
Wuyiencin EC_50_ + Pyrimethanil EC_50_ (7:3)	0.70 ± 0.24 b	51.05	25.93 ± 1.61 d	66.06

Note: Each datum is the mean of 3 replicates with standard error and different letters after the data indicate that they are significantly different at *p* < 0.01 tested with Duncan’s new multiple range test method.

**Table 5 microorganisms-12-01383-t005:** Classification of resistance levels of different sources of strains of gray mold.

Strain	Virulence Regression Curve (y=)	EC_50_ (μg mL^−1^)	R^2^	95% Confidence Interval	Resistance Coefficient RR	Resistance Level
57	0.54x−0.87	40.973	0.805	4.646–603.053	450.25	High resistance
59	1.95x−3.67	77.030	0.988	66.152–91.451	846.48	High resistance
61	1.47x−2.65	63.446	0.945	39.814–44.648	697.21	High resistance
65	0.92x−0.92	9.892	0.921	1.445–20.506	108.70	High resistance
503	0.42x + 0.07	0.565	0.753	0.001–2.871	6.208	Sensitive resistance
514	1.54x−1.92	17.329	0.928	3.021–37.097	190.43	High resistance

**Table 6 microorganisms-12-01383-t006:** Inhibition effect of pyrimethanil at different concentrations on different *B. cinerea* growth.

Strain	Fungicides Concentration (μg mL^−1^)	Disease Spot Diameter (cm)	Control Efficiency (%)
57	Control	7.18 ± 0.60 a	\
	5	4.68 ± 0.70 b	34.82
	25	4.61 ± 0.79 b	35.79
	50	3.23 ± 0.21 c	55.01
	100	3.09 ± 0.33 cd	56.96
	125	2.57 ± 0.14 b	64.21
59	Control	5.90 ± 0.35 b	\
	5	6.23 ± 0.82 a	−5.59
	25	4.81 ± 0.37 c	18.47
	50	3.97 ± 0.31 d	32.71
	100	2.48 ± 0.14 e	57.97
	125	1.89 ± 0.22 f	67.97
61	Control	6.53 ± 0.44 a	\
	5	6.05 ± 0.24 b	7.35
	25	5.33 ± 0.15 c	18.38
	50	3.83 ± 0.26 d	41.35
	100	2.42 ± 0.43 e	62.94
	125	1.32 ± 0.15 f	79.79
65	Control	6.81 ± 0.21 a	\
	5	3.83 ± 0.38 b	43.76
	25	3.10 ± 0.40 c	54.48
	50	1.66 ± 0.31 d	75.62
	100	1.20 ± 0.10 e	82.38
	125	0.90 ± 0.00 e	86.78
503	Control	5.90 ± 0.30 a	\
	5	1.95 ± 0.24 b	66.95
	25	1.60 ± 0.27 c	72.88
	50	1.58 ± 0.17 c	73.22
	100	1.15 ± 0.23 d	80.51
	125	0.75 ± 1.64 e	87.29
514	Control	7.47 ± 0.35 a	\
	5	5.78 ± 0.62 b	22.62
	25	3.14 ± 0.58 c	57.97
	50	2.60 ± 0.20 d	65.19
	100	0.58 ± 0.46 d	92.24
	125	0.00 ± 0.00 d	100.00

Note: The lesion diameter data are the means of 9 lesions with standard errors and different letters after the data indicate that they are significantly different at *p* < 0.05 analyzed with Duncan’s new multiple range test method.

**Table 7 microorganisms-12-01383-t007:** In vitro induction of highly resistant strains of *B. cinerea* to pyrimethanil.

Algebra	Index	Pyrimethanil EC_50_	Wuyiencin EC_50_ + Pyrimethanil EC_50_
1	Toxicity regression curve	y = 1.24x − 1.12	y = 1.26x − 0.85
	EC_50_ (µg mL^−1^)	8.354	5.153
	Resistance multiplicity	1	1
8	Toxicity regression curve	y = 2.29x − 1.76	y = 2.06x − 1.82
	EC_50_	6.086	7.610
	Resistance multiplicity	1.37	0.68
10	Toxicity regression curve	y = 0.77x + 0.13	y = 2.03x − 1.59
	EC_50_	0.663	6.008
	Resistance multiplicity	9.18	1.27

## Data Availability

The original contributions presented in the study are included in the article/Appendix A, further inquiries can be directed to the corresponding author.

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
