# Peer review of "Effect of Combining Wuyiencin and Pyrimethanil on Controlling Grape Gray Mold and Delaying Resistance Development in Botrytis cinerea"

_microorganisms, 2024, doi:10.3390/microorganisms12071383_

Round 1

Reviewer 1 Report

Comments and Suggestions for Authors

The current manuscript requires major revision before acceptance for publication. Below are the comments and questions:

Authors should check and follow the journal's instructions, using the SI measurement system for units in the manuscript. For example, use mg L-1 instead of mg/L.

Lines 22, 339: Italicized "B. cinerea"

Lines 34-37: Add related references to these sentences

Line 60: add related references to this sentence

Line 104: add related references to this sentence

In the materials and methods section: the authors should add the plant materials section because in the results section, the authors mentioned the effect of treatment on B. cinerea infection in grape leaves and fruits. Therefore, in the method materials section, the author must clearly state the plant materials used in the experiment, for example, the name of the grape variety, what leaf and fruit development stages were used in the experiment, the growth conditions, harvest time, characteristics of the grape cultivar used in the experiment (are they less susceptible, moderately susceptible, or highly susceptible to B. cinerea, etc.)

The experimental design was not reasonable: in the first basic experiment "Inhibitory effect of wuyiencin on Botrytis cinerea pathogenicity", the author determined that the concentration of 120 µg/mL wuyiencin was the most effective in suppressing the growth and development of B. cinerea. Therefore, in the next experiments, the author must choose this concentration for detailed investigation and evaluation. Meanwhile, the author chose a concentration of 100 µg/mL for "Infection pad formation", and chose 65 µg/mL to evaluate the expression of four pathogenicity related genes of Botrytis cinerea in grape leaf tissues,... An explanation is needed.

If higher concentrations of wuyiencin are used, how does it affect the growth of B. cinerea?

Figure 1, additional statistical analysis for control and wuyiencin data. Why was 65 µg/mL chosen for this experiment when 100 and 120 µg/mL were more effective?

BcSpl1, a cerato-platanin family protein, contributes to Botrytis cinerea virulence and elicits the hypersensitive response in the host, so in Figure 1, the authors should additionally evaluate the expression of the BcSpl1 gene.

Table S2: Add the accession number or reference for each gene, and clarify forward primers and reverse primers.

Figures 2, 3, and 4: Add a scale bar to each corresponding image

Figure S5: Adding statistical analysis to the data. On the X-axis, replace Chinese characters with English.

Comments on the Quality of English Language

Minor editing of English language required

Reviewer 2 Report

Comments and Suggestions for Authors

Botrytis cinerea is an ascomycetous fungus that causes great damage worldwide to various plant species in agriculture, horticulture and forestry. Chemical fungicides are commonly used to reduce damage. The current manuscript presents the results of research on the use of wuyiencin, a secondary metabolite produced by Streptomyces albulus var. wuyiensis, and its combination with chemical fungicides. The experiments are numerous, well thought out, and their results are as well documented as possible. These results led to specific findings and conclusions. These results are very valuable for science and practice, especially in the field of Vitis vinifera cultivation. The manuscript should be published in Microorganisms. Numerous corrections and additions should be made beforehand, some of them are very important (see Remarks).

Remarks

Line 22 B. cinerea – it should be in italic

Line 30 generally it is recommended that the Keywords should contain words other than those in the title

Line 41 mycelium or sclerotium germinated – please note that mycelium does not germinate (this term cannot be used)

Line 43 in the host tissue – it should be rather on the host tissue

Line 120, please provide details: composition, name of the company

Line 133 was examine on PDA plates - it should be ‘was examined on PDA plates’

Line 138 15 cm - diameter ?

Line 138 Infection pad formation: There are fundamental inconsistencies between the text in M&M and the text provided in the results as an explanation to Figure S4 (the authors contradict themselves);

a/ line 138 ‘sprayed with 0 (control), 50, 80, and 100 μg/mL of wuyiencin, respectively’ while Figure S4 : Line 114 Suppl. ‘effects of wuyiencin concentrations of 0 (control), 20, 60, and 100 μg/mL, respectively.

b/ Line 142-143 ‘and the leaves were inoculated with B. cinerea mycelial discs (5 mm in diameter) at 24 h after wuyiencin treatment’ while Figure S4 Line 115 Suppl ‘Leaves were inoculated 24 h before wuyiencin treatment’

Line 137 How many leaves are placed in one Petrie dish?

Line 137-139 consider revising this sentence

Line 142 one mycelial disc for one leaf?

Line 144 how many leaf pieces were taken from one inoculated leaf

Line 195  you should specify how many leaves and how many fruits were put into one Petrie dish and how many were analyzed (this data should be written in detail in the methods)

Line 215 was there a control?

Line 297-301- consider revising this text

Line 316 it should be Johnson instead of Jonson

Line 320 and other places in text - Botrytis cinerea should be in italic

Line 354 – 358 consider revising this text

Line 390 at160 rpm – add a space

Line 484-487 This text should be in Discussion, not in Result.

Line 532 'saprophytic tissue' - this term must be changed (probably it means dead tissue?)

Line 586 it should be Scanning

Line 588 You demonstrate in Figure 3 and in the text (line 224, 565) that B. cinerea produces conidia inside leaf tissue. It is necessary to refer to this fact in the discussion - is this a known phenomenon in the development cycle of B. cinerea, or are your observations made for the first time? It should be stated whether the conidia inside the plant tissue are identical to those produced outside the plant tissue. Sometimes the fungi produce different yeast-like conidia inside the plant tissue than outside the tissue. Relevant literature should be cited in the Discussion. Currently, there is no information about this incident in the discussion.

Line 603 consider revising this text

Line 614 the text needs to be moved to another place

Line 642 51.23 , Line 640 290.59 – compare with data in Table 1

Line 681 it should be Johnson instead of Jonson

Line 681 it should be 1960 instead of 1950. Also, the number should be given [41]

Line 814-815 it should be biosynthesis

Line 842 Pseudomonas - italic

Line 843 You use the name Lactobacillus plantarum, while in the cited paper [55] the name Lactiplantibacillus plantarum is given. This requires explanation.

Line 945 it should be Botrytis cinerea Pers. instead of Botrytis cinernea pers.

Line 983 Pseudomonas fluorescens – italic

Line 991 it should be Botrytis instead of Botryris

Line 1044 plantarum - italic

Line 1054 Alternaria panax - italic

Supplement

Line 35 add 'and conidia germination'

Line 52 9lesions ?

Line 69 after how many days the colony diameter was measured

Reviewer 3 Report

Comments and Suggestions for Authors

Title

Title does not indicate the actual content in manuscript. Modify it. See comments in MS.

Abstract

It needs an introduction and to point out the problem.

Use as keywords significant words but not the ones in title

Materials end methods

In general, this section is well written, however, information in lines 112 to 123 needs more detailed information. It reads: “The grape cultivars Rosescent (for seedling samples) and Fujiminori (for fruit samples) were grown in a greenhouse at the IPP-CAAS”. Seedling refers for young plants, but in the information given those young plants never appear. Leaves and fruits are used for the experiments.

Results

At the beginning of the section figures and tables in complementary documents are refer. Why no to include those in the manuscript?

Discussion and conclusion

This discussion is too short taking into account all the experiments and results presented. Finally, only the combination wuyiencin and pyrimethanil is consider for discussion, what about the other chemicals in the study?

General comment

The information presented in the manuscript is too large. I refer to the experiments and so the results, they are many experiments and sometimes the chronological or logical development are not well explained. They need better order to be understood. I feel like the more presented the less I understood.

Consider to write two manuscripts

See comments in MS.

Round 2

Reviewer 1 Report

Comments and Suggestions for Authors

Accept in present form. However, there are several typo errors in the Supplementary file. Authors should check and revise. 

Author Response

Answer to the specific questions/comments:

According to the reviewers’ comment, the manuscript has been revised as following.

Reviewer:

  1. Accept in present form. However, there are several typo errors in the Supplementary file. Authors should check and revise. 

Our response: Thanks for reviewer’s careful checking. We have modified.

Reviewer 3 Report

Comments and Suggestions for Authors

One of my comments in the first round of revision was to modify the title. I said: That the combination of biocontrol with chemical fungicides were the main focus of the paper. So, I suggested to change the title.

Author´s response indicate that changes in title was carried out, however titles in first and second versions of the MS are the same.

Author Response

Answer to the specific questions/comments:

According to the reviewers’ comment, the manuscript has been revised as following.

Reviewer:

  1. One of my comments in the first round of revision was to modify the title. I said: That the combination of biocontrol with chemical fungicides were the main focus of the paper. So, I suggested to change the title. Author´s response indicate that changes in title was carried out, however titles in first and second versions of the MS are the same.

Our response: Sorry, the title of the article was not corrected in time last time probably due to an input error. We have revised the title through discussion to “Effect of wuyiencin combined with pyrimethanil on the control of grape gray mold and its mechanism of delaying resistance development in Botrytis cinerea”. If you have any other comments, please provide guidance as well, thank you.